# Electrochemical Synthesis of Nano-Sized Silicon from KCl–K₂SiF₆ Melts for Powerful Lithium-Ion Batteries

**Timofey Gevel** [1,2], **Sergey Zhuk** [1,2], **Natalia Leonova** [1], **Anastasia Leonova** [1], **Alexey Trofimov** [1,2], **Andrey Suzdaltsev** [1,2,*] **and Yuriy Zaikov** [1,2]

1 Scientific Laboratory of the Electrochemical Devices and Materials, Ural Federal University, Mira St. 28, 620002 Yekaterinburg, Russia; Timofey.gevel@urfu.ru (T.G.); zhuksi83@mail.ru (S.Z.); n.m.leonova@urfu.ru (N.L.); a.m.leonova@urfu.ru (A.L.); a.a.trofimov@urfu.ru (A.T.); i.p.zaikov@urfu.ru (Y.Z.)
2 Institute of High-Temperature Electrochemistry, Ural Branch, The Russian Academy of Sciences, Academicheskaya St. 20, 620137 Yekaterinburg, Russia
* Correspondence: suzdaltsev_av@ihte.uran.ru or a.v.suzdaltsev@urfu.ru

**Abstract:** Currently, silicon and silicon-based composite materials are widely used in microelectronics and solar energy devices. At the same time, silicon in the form of nanoscale fibers and various particles morphology is required for lithium-ion batteries with increased capacity. In this work, we studied the electrolytic production of nanosized silicon from low-fluoride KCl–K₂SiF₆ and KCl–K₂SiF₆–SiO₂ melts. The effect of SiO₂ addition on the morphology and composition of electrolytic silicon deposits was studied under the conditions of potentiostatic electrolysis (cathode overvoltage of 0.1, 0.15, and 0.25 V vs. the potential of a quasi-reference electrode). The obtained silicon deposits were separated from the electrolyte residues, analyzed by scanning electron microscopy and spectral analysis, and then used to fabricate a composite Si/C anode for a lithium-ion battery. The energy characteristics of the manufactured anode half-cells were measured by the galvanostatic cycling method. Cycling revealed better capacity retention and higher coulombic efficiency of the Si/C composite based on silicon synthesized from KCl–K₂SiF₆–SiO₂ melt. After 15 cycles at 200 mA·g$^{-1}$, material obtained at 0.15 V overvoltage demonstrates capacity of 850 mAh·g$^{-1}$.

**Keywords:** lithium-ion battery; silicon; halide melt; electrodeposition; silicon fibers; nanotubes; nanoneedles; composite anode

## 1. Introduction

Currently, silicon and silicon-based composite materials are widely used in microelectronics, solar energy devices, as well as in the manufacture of portable energy storage devices [1–3]. In particular, a new lithium-ion battery (LIB) with a silicon-based anode is being actively developed. The theoretical capacity of such an anode (4200 mAh·g$^{-1}$) is an order of magnitude higher than the widely used graphite anode [3,4]. However, during cycling, silicon is subject to huge volumetric expansion (up to 400% [5]), which leads to mechanical destruction of the electrode and significant loss of the original capacity. To avoid this, it is recommended to use anodes based on nanosized silicon particles. Nanosized silicon cannot be obtained by the existing methods of the carbothermal reduction of quartz to silicon with subsequent purification of the silicon by hydrogenation–dehydrogenation [6]. The use of nanosized silicon particles, mainly fibers and tubes within composite materials, can significantly reduce the problem of volumetric expansion. Suitable polymer binders also reduce mechanical degradation of the electrode, but they do not affect the silicon expansion [7,8]. Another method that can improve silicon's performance is the coating (mainly carbon based) of silicon particles and creation of a "core-shell" structure [9,10]. Doped silicon anodes have also shown improved electrochemical performance and cyclic resistance in comparison with pure silicon [11–13].

Methods for preparing silicon nanofibers can be divided into two groups. The first group comprises methods based on removing excess material-laser ablation [14], ion etching [15], chemical etching [16,17]. These methods allow one to obtain high-purity components with the required sizes. However, they require expensive equipment, catalysts (gold is most often used) and are poorly suited for industrial scale production. The second group of methods includes silicon deposition on the substrates. Among these, the most commonly used method is the vapor–liquid–crystal deposition [18,19]. Silane is usually used as a source of silicon. The disadvantages of these methods also include the use of gold as a catalyst and the difficulty in scaling and hardware design of the process. To obtain silicon suitable for microelectronics, several stages of purification are needed, one of which is represented by a group of methods based on zone crystallization from molten silicon [20].

One of the cheapest and promising methods for producing nanosized silicon is electrolytic reduction from molten salts. To date, many studies have been carried out aimed at the electrolytic production of silicon from various molten electrolytes with $K_2SiF_6$, $Na_2SiF_6$, $SiO_2$, and $SiCl_4$ additives in a temperature range mainly from 550 °C to 750 °C [21–39]. All these studies show the fundamental possibility of using electroreduction for the production of silicon deposits of various morphologies, including continuous coatings up to 1 mm thick, submicron films (0.5–1 μm), micro-sized dendrites, nano- and micro-sized disordered fibers.

One of the most promising electrolytes for silicon production is the KCl–KF melt with a KF molar fraction of up to 66% [27–31], which is a good solvent for $K_2SiF_6$ and $SiO_2$ and is water soluble. However, this system also has a number of disadvantages, including relatively high aggressiveness of KF to reactor materials, the need to remove impurities such as $H_2O$ and HF from KF when preparing a molten KCl–KF mixture, and a good solubility of oxides. Despite the positive results of the work performed, all of the above factors can impede the deposition of high-purity silicon, especially without oxygen inclusions.

To eliminate these drawbacks, alternative media with the reduced or zero fluoride content for the production of silicon are being investigated. In particular, this concerns the KCl–KI–KF–$K_2SiF_6$ [32,33] and CaCl$_2$–CaO–SiO$_2$ melts [34–37], although their use also implies the absence of moisture in the reactor and increased requirements for the preparation of molten electrolytes and instrumentation. In previous works, we have shown the fundamental possibility of obtaining nanosized silicon fibers by electrolytic refining of silicon in a low-fluoride KCl–$K_2SiF_6$ system [38,39]. The advantage of this system is that its anionic composition virtually does not change. That provides stabilization of the energy characteristics of silicon-containing electroactive ions and possibility of controlling the morphology of silicon deposits. Preliminary electrolysis tests [38,39] show that it is difficult to obtain nanosized silicon deposits from molten salts. In particular, this is due to the varied concentration of silicon-containing ions in the melt, the semiconducting nature of silicon, and increased sensitivity of the deposit morphology to oxygen impurities.

Despite the active interest in Si-based anodes for LIBs [3,4,6–19,40], the works devoted to the utilization of electrolytic silicon deposits as the LIB anode material are limited [41–46]. The highest characteristics for silicon obtained by electrodeposition are on carbon cloth. The capacity was more than 1100 mAh·g$^{-1}$ by the 200th cycle at a discharge current of 500 mA·g$^{-1}$, while the silicon nanopowder itself, without the use of carbon fabric, demonstrates very low capacity—less than 500 mAh·g$^{-1}$ by the 5th cycle and less than 200 mAh·g$^{-1}$ by 200th cycle [40]. In recent years, most researchers are focused on silicon electrodeposition from CaCl$_2$-based melts. In work [42] carried out by researchers from the USA and China, electrodeposited nanofibers in LIB demonstrate a capacity of 714 mAh·g$^{-1}$ after 500 cycles at C/2 current. According to [45], China launched a large-scale test production of electrodeposited silicon with capacity more than 2000 mAh·g$^{-1}$ after 70 cycles at a current of 0.1 C; capacity at a current of 5 C is about 1600 mAh·g$^{-1}$. It is

noted that for practical use of the produced silicon they made Si/graphite mixture with a total capacity of 650 mAh·g$^{-1}$ [47].

In this regard, in this work, we studied the effect of $SiO_2$ on the morphology of nanosized silicon deposits during the electrolysis of KCl–$K_2SiF_6$ melts. For this, silicon deposits were obtained from the KCl–$K_2SiF_6$ and KCl–$K_2SiF_6$–$SiO_2$ melts at different cathode overvoltage. Obtained deposits were analyzed and tested on their electrochemical characteristics as a part of an Si-based anode of a LIB.

## 2. Materials and Methods

*Melt preparation.* The investigated KCl electrolytes with additions of $K_2SiF_6$ and $SiO_2$ were prepared by mixing individual salts of a chemically pure grade (Reakhim, Russia) and then melting them in a glassy carbon crucible immediately before the experiments. In a previous study [48], it was shown that the purity of KCl used is close to that of the recrystallized salt; therefore, additional purification of the prepared melt from electropositive impurities was not carried out. The $K_2SiF_6$ salt was preliminarily purified from oxygen impurities by HF fluorination. For this, $K_2SiF_6$ was mixed with $NH_4F$ and heated in stages to a temperature of 450 °C according to a previously described procedure [49]. $SiO_2$ was added to the melt in an amount of 0.34 wt.%. The KCl–$K_2SiF_6$ and KCl–$K_2SiF_6$–$SiO_2$ melts were held for an hour at the operating temperature to establish equilibrium between silicon ions [50,51]. Then, the melt samples were taken for analysis, and electrolysis was started.

*Experimental setup.* Electrolysis tests were carried out in a sealed quartz retort with a high-purity argon atmosphere at a temperature of 790 °C (Figure 1). A glassy carbon crucible with the melt was placed at the bottom of the retort, which was hermetically closed with a fluoroplastic cap. The inner walls of the retort were additionally shielded with nickel plates to protect against fluorine-containing sublimates. Holes with fittings were made in the cap, in which a working electrode (glassy carbon), a glassy carbon counter electrode, and a silicon QRE were placed. In addition, holes were provided for an additional working electrode, thermocouple, and a tube for gas supply and loading of silicon-containing additives. During the experiments, the temperature was controlled within ±2 °C by a Pt/PtRh thermocouple and a thermocouple module USB-TC01 (National Instruments, Austin, TX, USA).

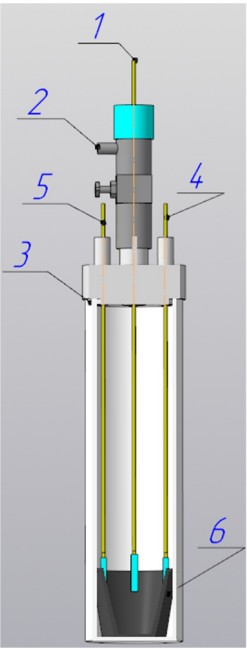

**Figure 1.** Experimental setup. 1—working electrode; 2—gas inlet; 3—quartz cell; 4 —counter electrode; 5—quasi-reference electrodes; 6—glassy carbon crucible.

*Electrochemical measurements and electrodeposition.* To establish the effect of $SiO_2$ on the kinetics of silicon electrodeposition, voltammograms were obtained by the cyclic chronovoltammetry. Before measurement, the electrodes were kept in the melt for 30 min to establish a stable (within $\pm$ 5 mV) potential difference between the working electrode and the quasi-reference electrode (QRE). In order to determine and compensate the ohmic voltage drop in the measuring circuit, impedance spectroscopy and current interruption (I-Interrupt) technique were used.

The electrodeposition of silicon was carried out on glassy carbon plates with an area of 2 $cm^2$ at different cathode overvoltage for one hour. The electrolysis time was selected based on the results of preliminary electrolysis tests [24,25]. For electrochemical measurements and electrolysis, PGSTAT AutoLAB 302N with Nova 1.11 software (MetrOhm, Herisau, Switzerland) was used. Glassy carbon plates were used as anodes.

At the end of the electrolysis, the electrode was raised above the melt for 20–30 min in order to remove excess salts included in the cathode deposit. Then, the electrode was raised to the cold zone of the retort and, after cooling in an inert atmosphere, was removed.

*Separation of the cathode deposit.* The obtained silicon deposits were washed in acidic (HCl) water solution; the pH of the solution for washing varies in the range of 2 to 4. An ultrasonic homogenizer SONOPULS UW mini 20 was used to disperse the deposits. Dispersion was carried out in a periodic mode at a given power of 0.995 kJ with a pulse duration of 90 s. After washing, the deposits were dried in a vacuum oven at 80 °C for 18 h.

*Analysis of the morphology and composition.* The elemental composition of the melt and the obtained silicon deposits was analyzed by atomic emission spectroscopy with inductively coupled plasma (AES-ICP) using an iCAP 6300 Duo Spectrometer (Thermo Scientific, Waltham, MA, USA). The morphology and elemental composition of the obtained deposits were studied using a Tescan Vega 4 (Tescan, Kohoutovice, Czech Republic) scanning electron microscope with an Xplore 30 EDS detector (Oxford, UK).

*Electrochemical performance.* Silicon performance was investigated in a 3-electrode half-cell. The anode composition was 10 wt.% polyvinylidene difluoride dissolved in N-methyl-2-pyrollidone, 10 wt.% carbon black, and 80 wt.% silicon. LIB fabrication was performed in an argon-filled glove box ($O_2$, $H_2O$ < 0.1 ppm). Stainless steel mesh with applied composite anode was used as the working electrode and two separate lithium strips as the counter and reference electrodes. All electrodes were divided by two layers of separator and tightly placed in the cell. The cell was flooded with 1 mL of electrolyte—1 M $LiPF_6$ in a mixture of ethylene carbonate/dimethyl carbonate/diethyl carbonate (1:1:1 by volume). Cycling experiments were performed using a Zive-SP2 potentiostat (WonATech, Seoul, Korea).

## 3. Results

### 3.1. Electrolysis Test Results

According to ICP, the silicon contents in the $KCl$–$K_2SiF_6$ and $KCl$–$K_2SiF_6$–$SiO_2$ melts before electrolysis tests were 0.05 and 0.15 wt.%, respectively. The deposits were obtained at cathode overvoltage of 0.1, 0.15, and 0.25 V relative to the QRE potential. For the $KCl$–$K_2SiF_6$ melt (0.05 wt.% of silicon), the obtained deposits are shown in Figure 2. It can be seen that the growth of deposits occurs with the formation of silicon-salt mixtures. In this case, with a decrease in the cathode overvoltage, the deposits change color from gray to light brown. This can be associated both with a change in the crystalline structure of silicon to amorphous, and with a change in the size of silicon particles [31,38].

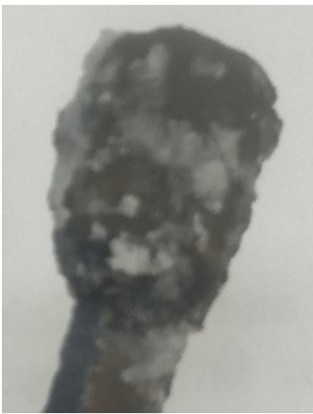
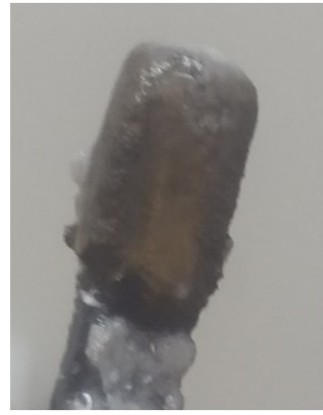
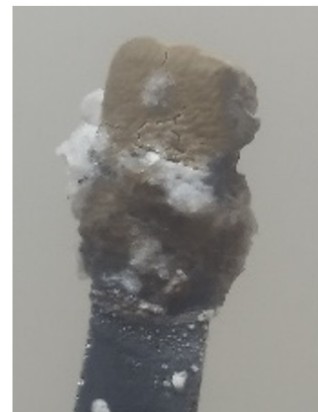

**Figure 2.** Photographs of the cathode deposits obtained after the electrolysis of the KCl–K$_2$SiF$_6$ melt at a temperature of 790 °C and cathode overvoltage of 0.25, 0.15, and 0.1 V vs. QRE potential.

Electrolysis tests in the KCl–K$_2$SiF$_6$ melt with the addition of SiO$_2$ (0.15 wt.% of silicon) at the same cathode overvoltages of 0.1, 0.15, and 0.25 V vs. QRE potential were performed. As a result, cathodic deposits were obtained, photographs of which are shown in Figure 3. It can be noted that during electrolysis of the melt with the addition of SiO$_2$, more volumetrically uniform deposits are obtained, while the color of the precipitates remains the same in all tests.

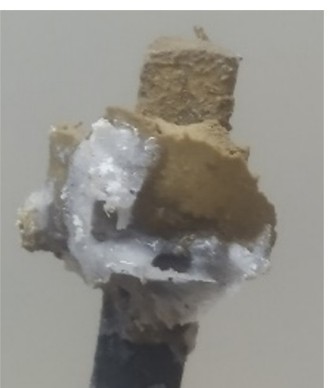
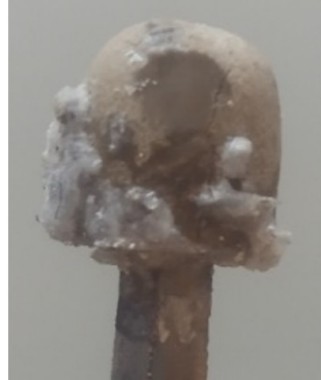
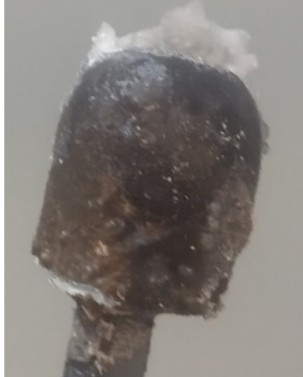

**Figure 3.** Photographs of the cathode deposits obtained after the electrolysis of the KCl–K$_2$SiF$_6$–SiO$_2$ melt at a temperature of 790 °C and cathode overvoltage of 0.25, 0.15, and 0.1 V vs. QRE potential.

### 3.2. Effect of SiO$_2$ on the Morphology of Silicon Deposits

For more detailed analysis of the effect of SiO$_2$ on the morphology of the silicon, SEM images were made. Figure 4 shows the corresponding micrographs of the deposits. It can be seen that in the KCl–K$_2$SiF$_6$ system, silicon was deposited in the form of disordered fibers with an average diameter of 300–400 nm, while the effect of cathode overvoltage on the morphology of the deposit is not clearly observed. Such a result is probably due to the limiting stage of the preceding chemical reaction of silicon-containing ion dissociation.

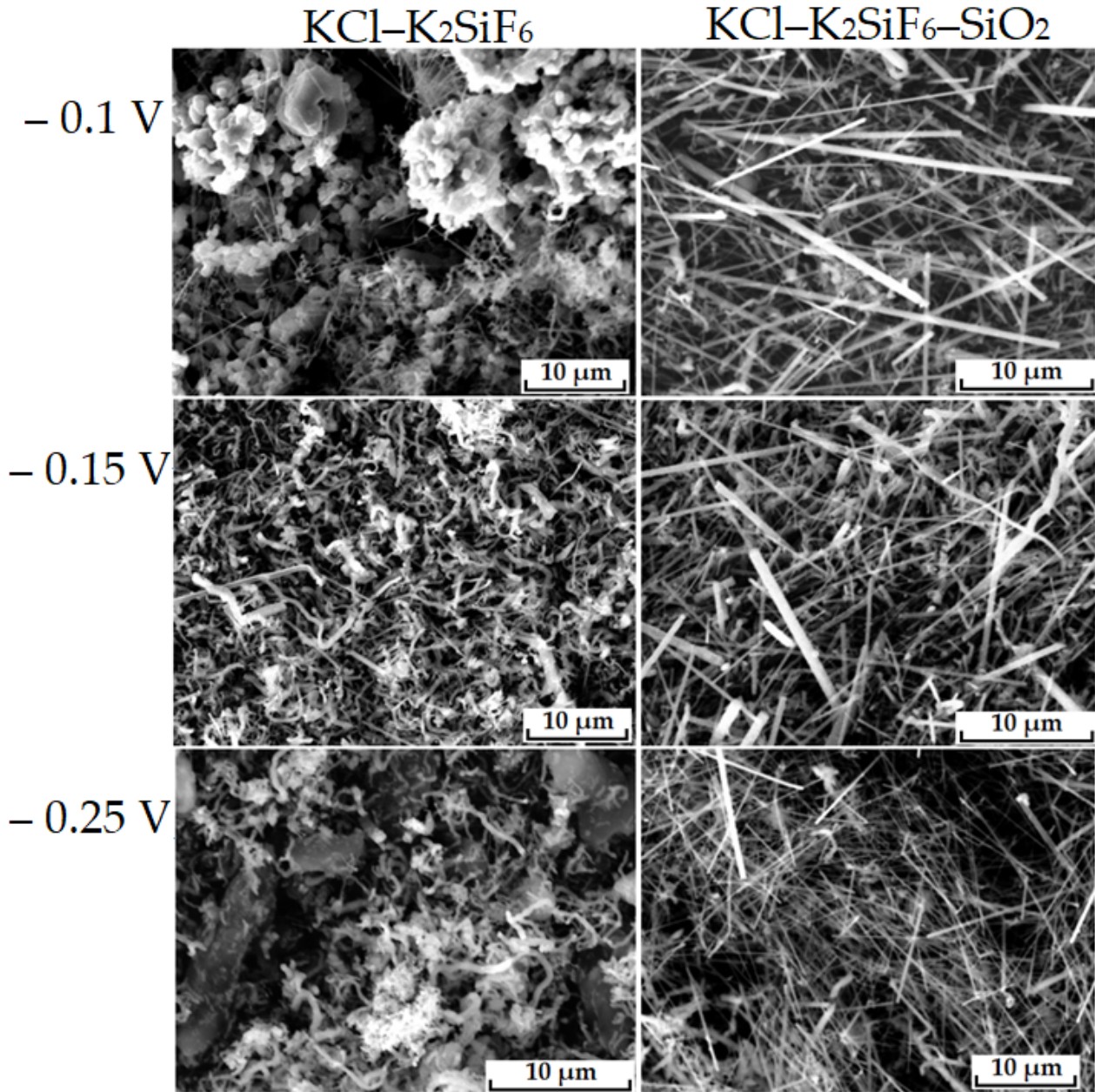

**Figure 4.** Micrographs of the cathode deposits obtained during the electrolysis of KCl–K$_2$SiF$_6$ and KCl–K$_2$SiF$_6$–SiO$_2$ melts at a temperature of 790 °C and cathode overvoltage of 0.25, 0.15, and 0.1 V vs. QRE.

When SiO$_2$ is added in the melt, silicon is predominantly deposited in the form of nano-sized tubes and needles, the length of which varies in the range of 30–60 μm, and the diameter varies in the range of 200 nm to 400 nm. This effect can be interpreted as the mutual influence of several factors:

(1)  a change in the composition of silicon-containing electroactive particles, namely the inclusion of oxygen atoms in their composition, leading to a decrease in binding energy of silicon atoms [51] and facilitation of the charge transfer stage;

(2)  a decrease in electrical conductivity (increase in resistance) of the near-cathode layer melt, which will redistribute the current lines among the cathode surface. Namely, silicon electrodeposition will occur at the tips of the deposit, rather than on the lateral surfaces;

(3)　a change in the composition of silicon-containing electroactive ions, as a result the ions activity and the rate constant of the possible preceding chemical reaction in the melt change.

Taking into account the weak effect of the cathode overvoltage on the morphology of silicon deposits, the effect of $SiO_2$ addition can be associated with the last two factors. It can be noted that the average size of silicon particles decreases with the addition of $SiO_2$ to the melt. A similar effect was observed in other works [31,41].

According to EDX analysis, the average oxygen content in the fibers obtained during the electrolysis of the $KCl$–$K_2SiF_6$ melt was 2.1–4.3 wt.%, and the average oxygen content in nano-sized tubes and needles obtained during the electrolysis of this melt with the addition of $SiO_2$ was 2.7–5.1 wt.%. The appearance of oxygen in the deposits can be caused both by the use of a quartz retort as a reactor container and after-electrolysis silicon treatment. It is important to note that the addition of $SiO_2$ does not significantly increase oxygen content in silicon deposits, which is important from the point of the electrochemical characteristics as the anode of LIB.

### 3.3. Effect of $SiO_2$ on the Kinetics of Silicon Electrodeposition

To check the effect of $SiO_2$ addition on the silicon electroreduction, voltammograms were recorded in the $KCl$–$K_2SiF_6$ and $KCl$–$K_2SiF_6$–$SiO_2$ melts at a temperature of 790 °C. Figure 5 shows that the electroreduction of silicon-containing electroactive ions from the $KCl$–$K_2SiF_6$ melt occurs at a potential more negative than 0.1 V, with the formation of a cathodic peak **Si** [27,32] at a potential of –0.1 V, relative to the QRE potential (Figure 5, black line).

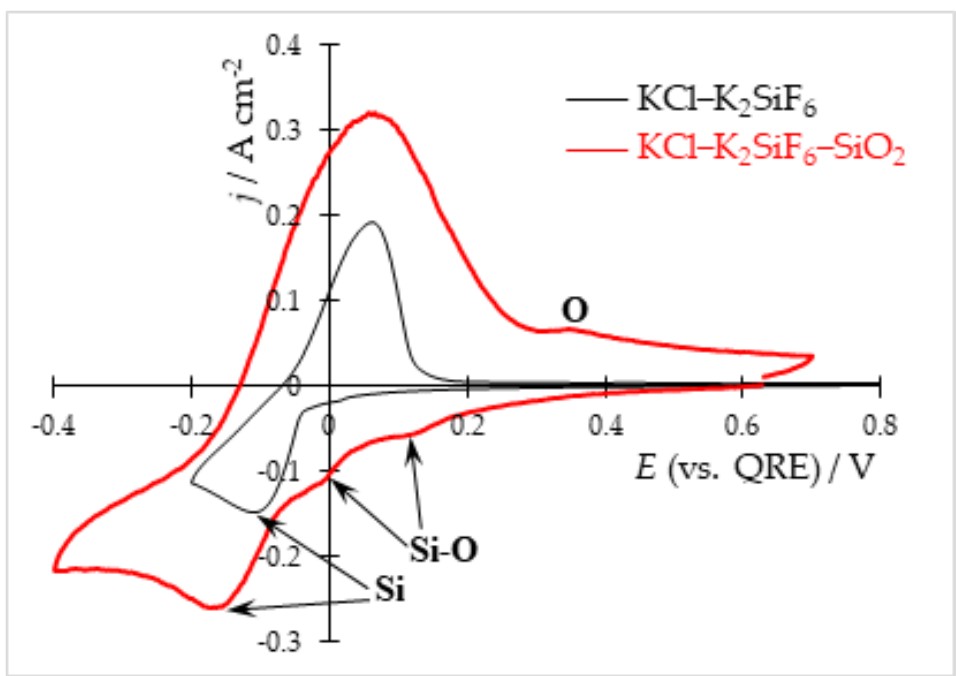

**Figure 5.** Cyclic voltammograms for the $KCl$–$K_2SiF_6$ and $KCl$–$K_2SiF_6$–$SiO_2$ melts with a total silicon content of 0.05 and 0.15 wt.%, respectively. Potential sweep rate is 0.4 V s$^{-1}$.

For the $SiO_2$-containing melt, additional **Si-O** cathode waves appeared on the voltammogram at the potentials of about 0.1 and 0 V relative to the QRE potential (Figure 5, red line). Those waves can be associated with the discharge of different oxygen-containing electroactive Si-O-Cl-F anions, which appear in the melt during $SiO_2$ dissolution [50,51]. A similar effect was observed in the works, aimed at the electroreduction of silicon and other elements from halide-oxide melts [52–54]. In the $KCl$–$K_2SiF_6$–$SiO_2$ melt, an increase of the silicon electroreduction current was observed, since the silicon content has been

increased. In the anode region, peak **Si′** and a wave **(Si-O)′** at a potential of 0.1 and 0.33 V relative to the QRE potential were detected. We assume that the peak **Si′** is associated with silicon dissolution [27,32], and the additional wave **(Si-O)′** can be caused by oxidation of a reduced form of the silicon electroactive ion. To obtain a more detailed picture, further study of the cathodic process in $KCl–K_2SiF_6$ and $KCl–K_2SiF_6–SiO_2$ melts is required.

### *3.4. Electrochemical Characteristics of the Obtained Silicon Deposits*

The obtained silicon deposits were studied as a composite Si/C anode for a lithium-ion battery. In the first charge/discharge cycle (Figure 6), the capacity of Si/C composite prepared on the base of silicon electrodeposited from $KCl–K_2SiF_6$ was between 1450 and 3000 mAh·$g^{-1}$ during lithiation, and between 950 and 1670 mAh·$g^{-1}$ during delithiation. Despite of the similar morphology and size of silicon fibers synthesized at different overpotentials, their energy characteristics are different. As can be seen from Figure 6, samples 2 and 3 synthesized at 0.15 and 0.25 V show similar charging times, although the initial period associated with electrolyte reduction and solid electrolyte interphase formation is quite different. Silicon synthesized at 0.1 V demonstrates 60% lower capacity, and the highest initial coulombic efficiency of 65.5%.

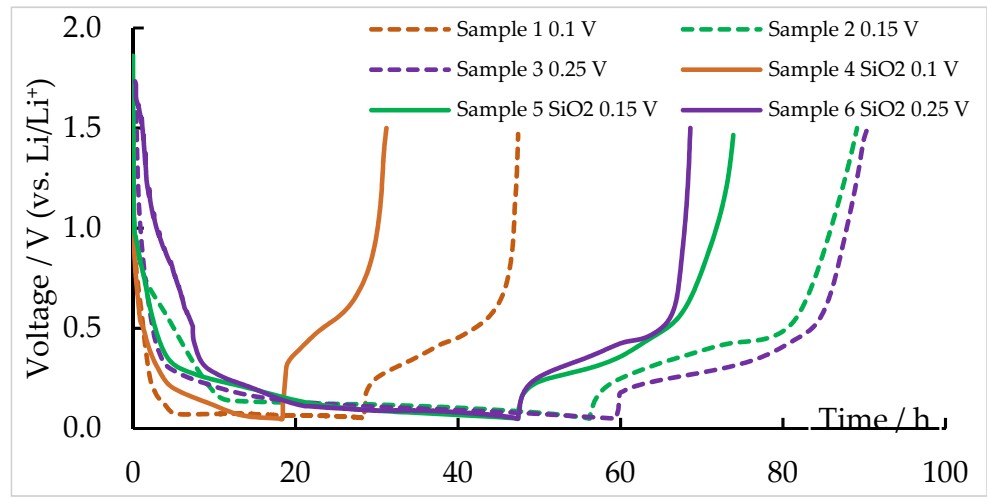

**Figure 6.** First charge/discharge cycle curves of Si/C composites based on electrodeposited silicon at 50 mA $g^{-1}$. Solid lines represent silicon deposited from molten $KCl–K_2SiF_6$; dashed lines represent silicon deposited from molten $KCl–K_2SiF_6–SiO_2$.

After 15 cycles at 200 mAh·$g^{-1}$ (Figure 7a), the capacitance for the best sample of silicon fibers was 750 mAh·$g^{-1}$ for sample 3 synthesized at 0.25 V overvoltage. The coulombic efficiency of the anode half-cell with silicon fibers in this case increased from 59% up to 92%. Silicon synthesized at 0.15 V (sample 2) demonstrates a similar capacity fade rate as sample 3, but a lower overall capacity. However, these samples have different coulombic efficiency during the first 10 cycles. Sample 1, while having a much lower capacity, demonstrates better capacity retention, with −50.7% of the initial capacity after 15 cycles.

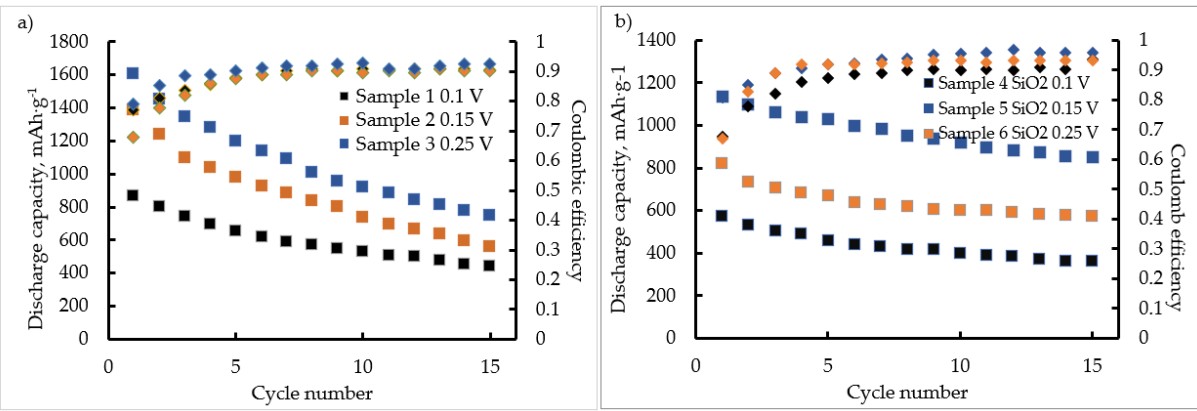

**Figure 7.** Cycling performance and the coulombic efficiency of the Si/C made from silicon electrodeposited from (**a**) KCl–K$_2$SiF$_6$ (samples 1–3) and (**b**) KCl–K$_2$SiF$_6$–SiO$_2$ (samples 4–6) at the cathode overvoltages of –0.1, –0.15, and –0.25 V. Diamonds-coulombic efficiency; squares-capacity.

For silicon samples obtained from the KCl–K$_2$SiF$_6$–SiO$_2$ melt, the anode capacity was between 920 and 2430 mAh·g$^{-1}$ for charging and between 600 and 1300 mAh·g$^{-1}$ during discharge. As can be seen from Figure 6, the behavior of these samples is similar to the ones synthesized without the addition of SiO$_2$ to the melt, but lower capacities are achieved for all the samples. The highest capacity value was observed for the sample 5 synthesized at 0.15 V overvoltage. The initial coulombic efficiency of the silicon needles/tubes was 44–65%, lower than for the samples 1–3. Sample 4 demonstrates much higher delithiation potential than all other samples, probably due to the higher C-rate in comparison to other samples. After 15 cycles (Figure 7b), samples synthesized from the melt with SiO$_2$ addition demonstrate higher coulombic efficiency—between 93.5% and 96.1%—and better capacity retention rate. Remaining capacity of the sample 5 is 850 mAh·g$^{-1}$, which is 74.5% of the initial capacity.

It is probable that such a difference in capacity fade rate is caused by different silicon structures or surface layer composition; however, the reasons for drastic capacity difference among the samples with similar morphology synthesized at different deposition overpotential is not clear. The results obtained indicate that further research is promising, aimed both at optimizing the conditions for obtaining silicon deposits from the investigated molten electrolytes, and at improving the performance of a lithium-ion battery with a composite anode based on the obtained silicon.

## 4. Conclusions

In this work, we studied the effect of cathode overvoltage and the addition of SiO$_2$ on the morphology of silicon electrolytic deposits obtained from KCl–K$_2$SiF$_6$ and KCl–K$_2$SiF$_6$–SiO$_2$ melts at a temperature of 790 °C. Using scanning electron microscopy, it was shown that during electrolysis of the KCl–K$_2$SiF$_6$ melt, nano-sized disordered fibers (300–400 nm in diameter) are formed on the cathode, while the addition of SiO$_2$ to the melt leads to an ordered growth of silicon deposit in the form of nano-sized needles and tubes (diameter 200–400 nm, length 30–60 μm). At the same time, it was noted that an increase in cathode overvoltage during the electrolysis of the studied molten electrolytes has virtually no effect on the morphology and size of the deposit, which may be associated with the course of the process under the conditions of a slowed-down preceding chemical reaction of the silicon-containing ion dissociation.

According to EDX analysis, the average oxygen content in silicon deposits obtained during the electrolysis of the KCl–K$_2$SiF$_6$ melt was 2.1–4.3 wt.%, and in the deposits obtained during the electrolysis of this melt with the addition of SiO$_2$, the average oxygen content was 2.7–5.1 wt.%.

The resulting deposits were used for the manufacture of composite Si/C anode half-cells of lithium-ion batteries. All the synthesized samples have similar initial charge/discharge curves, although the capacity is different. Samples synthesized at 0.1 V show much lower capacity than the ones synthesized at 0.15 and 0.25 V overvoltage. Although the first cycle capacity of silicon synthesized with $SiO_2$ addition is lower than that of silicon obtained without additive, further cycling results in higher capacity and much higher capacity retention for all samples. For silicon deposited from $KCl–K_2SiF_6$, the highest capacity value is 750 $mAh\cdot g^{-1}$ with capacity retention of 46.6 % and coulombic efficiency of 92.7%. The best results are achieved for silicon synthesized from $KCl–K_2SiF_6–SiO_2$ at 0.15 V overvoltage; its capacity was 850 $mAh\cdot g^{-1}$ after 15 cycles at a current of 200 $mA\cdot g^{-1}$ and capacity retention of 74.5% with coulombic efficiency of 96.1%. Such differences in the capacity retention for the samples synthesized with and without $SiO_2$ addition is not clear and requires further investigation.

**Author Contributions:** Conceptualization, T.G. and A.S.; methodology; T.G., S.Z., N.L., A.L. and A.T.; validation, T.G. and A.S.; formal analysis, A.S. and A.T.; investigation, T.G., S.Z., N.L., A.L. and A.T.; writing—original draft preparation, T.G. and A.S.; writing—review and editing, T.G. and A.S.; supervision, A.S. and Y.Z.; project administration, A.T. and A.S. All authors have read and agreed to the published version of the manuscript.

**Funding:** This research received no external funding.

**Institutional Review Board Statement:** Not applicable.

**Informed Consent Statement:** Not applicable.

**Acknowledgments:** This work is performed in the frame of the State Assignment number 075-03-2020-582/1, dated 18 February 2020 (the theme number 0836-2020-0037).

**Conflicts of Interest:** The authors declare no conflict of interest.

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
