# Peer review of "Electrochemical Synthesis of Nano-Sized Silicon from KCl–K2SiF6 Melts for Powerful Lithium-Ion Batteries"

_applsci, doi:10.3390/app112210927_

Round 1

Reviewer 1 Report

very interesting study with clear conclusion

beautiful nanofibers obtained with true potentialities

Author Response

Great thanks for the interest and kindly report!

Reviewer 2 Report

The manuscript gives a description of experiments on the electrochemical deposition of silicon from a KCl-K2SiF6 salt melt with and without the addition of SiO2. The manuscript includes characterization of the depositions for three different voltages for each type of melt, two U-I curves, and a verbal description of the results of electrochemical measurements in which the deposited silicon was used as an anode. The manuscript is one of a series of publications by the authors.

The article is written in a straightforward manner and is limited to a presentation of the experimental findings.  In its character, the manuscript is more of a technical report . Unfortunately, questions arising from the chosen approach are not addressed. For example, it would be interesting to discuss or refer to literature what effects the addition of SiO2 to the K2SiF6 melt has, for example, is SiF4 released or HF or similar.

The manuscript is suitable for publication, but a revision should be made to eliminate ambiguities and to increase the depth of information somewhat.

There are inconsistencies regarding the amount of SiO2 addition to the melt:

Page 2: line 81+82: "SiO2 81 was added to melt in an amount of 0.1 wt.%."

Page 4: line 151+152: "95KCl-5K2SiF6 melt with the addition of 0.34 wt.% SiO2".

Page 4: line 142+143:

"According to ICP, the silicon contents in the KCl-K2SiF6 and KCl-K2SiF6-SiO2 melts before electrolysis tests were 0.05 and 0.15 wt.%, respectively."

The silicon content calculated from the composition of the melt consisting of 5wt% K2SiF6 and 95wt% KCl should be much higher than the value determined in the ICP analysis. Please explain this significant difference.

Page 4: line 152: "the oxygen content in the system was 0.133 mol%"

How was the oxygen content in the melt determined? Please specify consistently in wt%. What is the significance of this value?

Fig.3: The scales of the SEM images are not readable, please use higher resolution images. Are all images selected at the same magnification?

It would be nice if more analysis results were shown, for example cross sections to show the morphology or EDX-mappings that allow conclusions about the homogeneity of the deposits.

Fig. 5: Please insert a name for the two curves (red and black) (legend)

Page 6: line 197 Section: 3.3 Effect of SiO2 on the kinetics of silicon electrodeposition

In this section U-I curves of Si deposition from SiO2-free and SiO2-containing melts are presented and characteristics of the curves are discussed. Explanations are given for the curves that cannot be provided from the context of the manuscript follow, for example "a clear cathodic peak Si at a potential of 0.1 V" or "additional Si-O cathode waves". Please cite the references for these interpretations.

In the title and abstract of the manuscript, reference is made to the electrochemical properties of the deposited silicon. In contrast, Section 3.4 only gives summary information on the electrochemical properties, however, it does not show any charge and discharge curves, capacitance curves at different charging rates, or the results of multiple measurements. If the manuscript already refers to the electrochemical properties in the title, this section should be provided with more detailed results.

Author Response

Please, find the file.

Reviewer 3 Report

Dear Authors,

Please cover the following comments:

1- Paper innovations should be highlighted at the end of introduction.

2- Literature review should be investigated more.

3- Bring the nomenclature with details.

4- English grammar should be improved.

Author Response

Please, find the file!
